# Body Roundness Index, A Body Shape Index, Conicity Index, and Their Association with Nutritional Status and Cardiovascular Risk Factors in South African Rural Young Adults

**DOI:** 10.3390/ijerph18010281

**Published:** 2021-01-01

**Authors:** Mbelege Rosina Nkwana, Kotsedi Daniel Monyeki, Sogolo Lucky Lebelo

**Affiliations:** 1Department of Physiology and Environmental Health, University of Limpopo, Polokwane 0727, South Africa; rosina.nkwana@yahoo.com; 2Department of Life and Consumer Sciences, University of South Africa, Florida 1710, South Africa; lebelol@unisa.ac.za

**Keywords:** cardiovascular risk factors, obesity, Body Roundness Index, A Body Shape Index, Conicity Index

## Abstract

Background: The study aimed to investigate the association of Body Roundness Index (BRI), A Body Shape Index (ABSI), and Conicity Index with nutritional status and cardiovascular risk factors in South African rural young adults. Methods: The study included a total of 624 young adults aged 21–30 years from the Ellisras rural area. Anthropometric indices, blood pressure (BP), and biochemical measurements were measured. Results: BRI was significantly correlated with insulin (0.252 males, females 0.255), homeostatic model assessment (HOMA)-β (0.250 males, females 0.245), and TG (0.310 males, females 0.216). Conicity Index was significantly associated with pulse rate (PR) (β 0.099, 95% confidence interval (CI) 0.017, 0.143, *p* < 0.013; β 0.081, 95% CI 0.000 0.130, *p* < 0.048), insulin (β 0.149, 95% CI 0.286 0.908, *p* < 0.001; β 0.110, 95% CI 0.123 0.757, *p* < 0.007). Conicity Index is associated with insulin resistance (IR) (odds ratio (OR) 7.761, 95% CI 5.783 96.442, *p* < 0.001; OR 4.646, 95% CI 2.792 74.331, *p* < 0.007), underweight (OR 0.023, 95% CI 0.251 0.433, *p* < 0.001; OR 0.031, 95% CI 0.411 0.612, *p* < 0.001), and obesity (OR 1.058, 95% CI 271.5 4.119, *p* < 0.001; OR 1.271, 95% CI 0.672 1.099, *p* < 0.001). Conclusion: Conicity Index was positively associated with insulin resistance, hypertension and dyslipidaemia. Further investigation of these indices and their association with nutritional status and cardiovascular diseases (CVDs) could assist in efforts to prevent CVD in the rural South African population.

## 1. Introduction

Anthropometric parameters are a reliable, non-invasive, and affordable measurement for cardiovascular diseases (CVDs) and their associated risk factors. However, some anthropometric measures are not effective in estimating abdominal obesity [1]. The World Health Organisation (WHO) established obesity using the anthropometry index as Body Mass Index (BMI) equal to 30 kg/m^2^ or greater [2]. However, this BMI has its limitation as it cannot reflect an individual’s fat distribution and cannot differentiate between fat mass and muscle weight. Therefore, waist circumference has been recommended as a substitute for the obesity index, which modulates the limitation of BMI [3].

Recently, two indices named Body Roundness Index (BRI) and A Body Shape Index (ABSI) were associated with the risk of premature death [4,5]. The A Body Shape Index was found to be related to abdominal adipose tissue [5]. It also suggested that a high ABSI value indicates a large accumulation of adipose tissue around the abdominal region and appears to be a significant risk factor for health complications [6]. Body Roundness Index determines visceral adipose tissue (VAT) and body fat percentage [4]. In addition to BRI, Thomas et al. [4] supported BRI to be the associated with premature death. Maessen et al. [7] firstly used the two indices to investigate their capability to be associated with CVDs and their associated risk factors. However, BRI and ABSI were not significantly associated with CVDs as compared to BMI and waist circumference (WC) [7]. 

One more form of the anthropometric index, i.e., the Conicity Index, was developed a few decades ago. The index was developed as an indicator of obesity and the distribution of body fat [8]. This index is an important clinical apparatus to be used to determine the risk of CVD in a population [1]. The Conicity Index is based on the hypothesis that individuals with more accumulation of fat around the abdomen have a double cone shape, whereas those who have less accumulation of fat around the central region have a cylinder shape. This index involves variables such as weight, height, and WC. Conicity Index and BRI were established as indicators of body fat distribution and their values increase according to the build-up of fats in the abdominal region of the body [4,9]. 

The relationships between anthropometric parameters, such as height, weight, Body Mass Index (BMI), and waist circumference (WC), and CVD risk factors have been established in longitudinal follow-up studies [10]. Hence, using these three indices (BRI, ABSI, and Conicity Index) in the study of premature death in rural population might be useful. Most studies used the two new indices with BMI and WC to determine the association with risk of CVDs [3,7]. However, in this study, we are using the two new indices (BRI, ABSI) and Conicity Index. The study aimed to investigate the association of BRI, ABSI, and Conicity Index with nutritional status and cardiovascular risk factors.

## 2. Materials and Methods 

### 2.1. Sample 

This study is part of the ongoing Ellisras Longitudinal Study (ELS), for which more details of the sampling procedure were reported previously [11]. However, this study, a cross-sectional study, included a total of 624 young rural South African adults (306 males, 318 females) from the Ellisras rural area. Data collection was carried out in November 2015. The exclusion criteria included pregnant women and those who failed to provide a consent form. Participants who were already taking diabetes, dyslipidaemia antihypertensive medication, and subjects who did not fast were also excluded because they will affect the blood results. Only participants who fell between the ages of 21 and 30 years without the characteristics mentioned above were included in this study. Participants signed informed consent forms. This ethical clearance for this study was approved by the University of Limpopo Turfloop Research Ethics Committee (TREC) (TREC/264/2019:PG). 

### 2.2. Data Collection

#### 2.2.1. Anthropometric Measurement 

Anthropometric measurements were carried out according to the International Society for the Advancement of Kinanthropometry [12]. Height was measured using a stadiometer to the nearest 0.1 cm with shoes and headbands off. Weight was measured using a digital standing scale to the nearest 0.1 kg with light clothing and no shoes. Waist circumference was measured using a non-stretchable measuring tape to the nearest 0.1 cm. Body Mass Index (BMI) was calculated using weight in kilograms divided by height in metres. All the measurements were done by trained personnel. Conicity Index was calculated using the formula below [8].
Conicity Index = (waist circumference (m))/(0.109 × √((body weight(kg))/(height(m))))(1)


A Body Shape Index was calculated using the following formula [4]:
ABSI = WC/[(BMI)^(2/3) × (height)^(1/2)](2)
and Body Roundness Index was calculated using the following formula [5]:
BRI = 365.2 − 365.5 × √(1 − (((wc/2π)2)/[(0.5 × height)]^2))(3)


#### 2.2.2. Blood Pressure and Pulse Rate (PR)

Blood pressure measurements were taken utilising an electronic Micronta monitoring kit. The systolic (SBP) and diastolic blood pressure (DBP) readings were taken three times at five-minute intervals. The device displays the BP and PR simultaneously on the screen. Normal BP for adults is 120/80 mmHg. Subjects with BP of ≥140/90 mmHg, were classified as hypertensive [13].

#### 2.2.3. Fasting Blood Sample

Subjects fasted for 8–10 h before blood collection. Registered nurses from Witpoort Hospital in Ellisras collected blood samples. The venous blood samples were collected in the morning, with the arm of the subject rested on a supportive prop during blood collection. 

#### 2.2.4. Fasting Blood Glucose (FBG) and Insulin Levels

Fasting blood samples were collected into 4 mL grey-top Vacutainer tubes containing sodium fluoride and oxalate for analysis of FBG, and 4 mL yellow-top Vacutainer tubes containing separation gel for analysis of fasting insulin. The collected blood was centrifuged for 10 min to obtain plasma and serum. The blood plasma was used to measure the level of FBG using the glucose oxidase method, on a Beckman LX20^®^ auto-analyser (Brea, CA, USA). The blood serum was used to measure insulin using the enzymatic assay kits on a Beckman LX20^®^ auto-analyser. 

Fasting blood glucose and fasting insulin were used to calculate homeostasis model assessment (HOMA)-IR and HOMA-β using the following formula [14]. HOMA-IR assesses insulin resistance, while HOMA-β assesses beta-cell function using fasting plasma insulin(FPI) and fasting plasma glucose(FPG).
HOMA-IR = [(FPI(mIU/L) + FPG(mmol/L))/22.5](4)
HOMA-β% = [(20 × FPI(mIU/L))/(FPG(mmol/L) − 3.5)](5)


#### 2.2.5. Total Cholesterol 

The total cholesterol (TC), triglyceride (TG) and high-density lipoprotein cholesterol (HDL-C) levels were measured by an enzymatic spectrophotometric technique. However, the procedure depended on a unique detergent that solubilises HDL-C lipoprotein particles only and releases HDL-C results. Low-density lipoprotein cholesterol (LDL-C) was calculated using the Friedewald formula [TC − (HDL-C + TG/2.2)] [15]. The enzymatic assay kits on a Beckman LX20^®^ auto-analyser were used to measure the serum lipid profile. All measurements were carried out with an AU480 Chemistry System from Beckman Coulter (Brea, CA, USA). All blood samples were analysed by the University of Limpopo Medical and Pathology laboratory staff.

### 2.3. Statistical Analysis

Descriptive statistics for Conicity Index, BRI, ABSI, SBP, DBP, PR, and biochemical blood parameters were reported. Independent t-test was used to compare the values of males and females. The participants were characterised as obese using BMI > 30 kg/m^2^ and underweight as BMI < 18.5 kg/m^2^ [16]. To characterise high fatness, BRI and ABSI of above 95th percentiles [17] and Conicity Index of >1.25 for males and >0.83 females [9] were used. Hypertension was defined as BP > 140/90 mmHg [18]. The International Diabetes Federation (IDF) [19] cut-off point was used to define diabetes and dyslipidaemia. The following cut-offs were set: high triglycerides ≥1.7 mmol/L, reduced HDL <1.03 mmol/L in males <1.29 mmol/L in females, high insulin >25 mlU/L, and high FBG ≥5.6 mmol/L. The homeostatic model assessment was classified as HOMA-IR ≥ 95th as insulin resistance and HOMA-β ≤ 5th as beta-cell dysfunction [20]. A larger value of HOMA-IR indicates higher IR, while a lower value of HOMA-B shows a β-cell dysfunction. Pearson correlation was performed to determine the correlation between anthropometric indices and cardiovascular risk factors. Linear regression coefficient analysis was used to assess the association between anthropometric indices and underweight, obesity and, cardiovascular risk factors, unadjusted and adjusted in accordance with participants’ age and gender. Logistic regression was used to estimate the odds of having CVD risk factors with high obesity indices, unadjusted and adjusted for age and gender. Conicity Index, BRI, and ABSI were used to determine their association with nutritional status and CVDs. The statistical significance was set at *p* < 0.05. All the statistical analyses of data were performed using the Statistical Package for the Social Sciences (SPSS) version 25 (IBM, Chicago, IL, USA). 

## 3. Results

As illustrated in Table 1 below, males show significantly (*p* < 0.05) higher mean values for both SBP and DBP (125.91 vs. 114.23 and 71.44 vs. 69.11, respectively) compared to females. Significantly, mean values for PR, Conicity Index, BRI, insulin, and LDL-C were higher in females (81.75 vs. 70.36, 0.90 vs. 0.84, 18.55 vs. 13.20, 10.86 vs. 7.75, and 2.73 vs. 2.35, respectively) than in males. HOMA-IR and HOMA-β means were significantly higher in females than in males (2.79 vs. 1.98; 1.98 vs. 2.76), respectively.

As depicted in Figure 1, males indicate higher prevalence for BRI, ABSI, SBP, and DBP (6.9% vs. 4.7%, 6.9% vs. 5.7%, 19.9% vs. 3.5%, and 5.9% vs. 4.4%, respectively) than females. On the other hand, females had higher percentages of Conicity Index, PR, FBG, insulin, HDL-C, LDL-C, and TC). The percentage for TG levels were almost the same for both genders (10.8% males, 9.1% females). Reduced HDL-C (24.4% males, 68.6% females) and higher FBG (44.1% males, 46.5% females) were the most prevalent, while high LDL-C (0.3% males, 1.9% females) was the least prevalent in both genders. 

As depicted in Figure 2, males indicated a higher prevalence of underweight (12.7%) than females (7.2%), on other hand females, showed a higher prevalence of overweight and obesity than males (23.3 vs. 9.5 and 24.8 vs. 4.6, respectively).

Table 2 indicates a correlation between Conicity Index and FBG (0.141 males, −0.185 females), insulin (0.143 males, 0.185 females), and TG levels (0.253 males, 0.237 females) which were not significantly different among genders except for insulin. Conicity Index was, furthermore, significantly associated with PR, BMI, and HDL-C (0.167, 0.150, and −0.253, respectively) in males. ABSI was only significantly associated with TG (−0.109 males, 0.120 females). Furthermore, there was a correlation between ABSI and LDL-C (0.152) in females and PR (0.148) in males. BRI was significantly correlated with insulin (0.252 males, females 0.255), HOMA-β (0.250 males, females 0.245), and TG (0.310 males, females 0.216). On another hand, HDL-C (−0.196) and BMI (0.144) were significantly associated with BRI in males only. There was no correlation between all anthropometric indices and both SBP and DBP in both genders. 

Table 3 shows the significant associations between anthropometric indices and various cardiovascular risk factors, unadjusted and adjusted for age and gender. Conicity Index was significantly associated with PR (β 0.099, 95% CI 0.017 0.143, *p* < 0.013; β 0.081, 95% CI 0.000 0.130, *p* < 0.048), insulin (β 0.149, 95% CI 0.286 0.908, *p* < 0.001; β 0.110, 95% CI 0.123 0.757, *p* < 0.007), HOMA-β (β 0.142, 95% CI 0.259 0.891, *p* < 0.001; β 0.103, 95% CI 0.096 0.740, *p* < 0.011), TG (β 0.146, 95% CI 0.147 0.486, *p* < 0.001; β 0.143, 95% CI 0.138 0.482, *p* < 0.001), and LDL-C (β 0.125, 95% CI 0.065 0.284, *p* < 0.002; β 0.081, 95% CI 0.003 0.225, *p* < 0.044). Triglyceride (β 0.090, 95% CI 0.022 0.314, *p* < 0.024), only was significantly associated with ABSI before adjusting the confounding factors. There was a significant association between BRI and DBP (β 0.113, 95% CI 0.000 0.002, *p* < 0.005; β 0.142, 95% CI 0.001 0.002, *p* < 0.001), insulin (β 0.152, 95% CI 0.005 0.014, *p* < 0.001; β 0.113, 95% CI 0.002 0.012, *p* < 0.005), TG (β 0.106, 95% CI 0.001 0.006, *p* < 0.008; β 0.112, 95% CI 0.001 0.006, *p* < 0.006), and TC (β 0.038, 95% CI 0.001 0.004, *p* < 0.001; β 0.112, 95% CI 0.001 0.003, *p* < 0.006) before and after adjusting with confounding factors for age and gender. 

As shown in Table 4, BRI is associated with dyslipidaemia and obesity before (odds ratio (OR) 2.721, 95% CI 1.307 5.665, *p* < 0.007; OR 3.607, 95% CI 2.911 4.471, *p* < 0.001) and after adjusting for age and gender (OR 2.138, 95% CI 1.010 4.526, *p* < 0.047; OR 3.557, 95% CI 2.841 4.454, *p* < 0.001), respectively. Furthermore, BRI is associated with underweight before adjusting for confounding (OR 0.201, 95% CI 0.127 0.316, *p* < 0.001). ABSI is associated with obesity (OR 0.019, 95% CI 0.004 0.095, *p* < 0.001; OR 0.123, 95% CI 0.022 0.675, *p* < 0.016) and underweight (OR 6.537, 95% CI 7.211 10.746, *p* < 0.001; OR 6.533, 95% CI 4.414 85.746, *p* < 0.002) before and after adjusting for confounding. Conicity Index is associated with IR (OR 7.761, 95% CI 5.783 96.442, *p* < 0.001; OR 4.646, 95% CI 2.792 74.331, *p* < 0.007), underweight (OR 0.000, 95% CI 0.000 0.000, *p* < 0.000; OR 0.000, 95% CI 0.000 0.000, *p* < 0.000), and obesity (OR 1.058, 95% CI 271.5 4.119, *p* < 0.001; OR 1.271, 95% CI 0.672 1.099, *p* < 0.001). Furthermore, Conicity Index is associated with dyslipidaemia before adjusting for confounding factors (OR 1.945, 95% CI 1.007 3.757, *p* < 0.048) and hypertension (HT) (OR 3.985, 95% CI 1.395 4.522, *p* < 0.029) after adjusting for co-founding factors.

## 4. Discussion

The study aimed to investigate the association between BRI, ABSI, and Conicity Index and nutritional status and cardiovascular risk factors in South African rural young adults aged 21−30 years old. In this cross-sectional study of rural South Africans, we found that Conicity Index had the strongest associations with CVD risk factors, while ABSI had the weakest associations. Body Roundness Index had intermediate associations with cardiovascular risk factors. Female young adults showed significantly higher mean values for PR, Conicity Index, BRI, insulin, and LDL-C. This put female young adults at higher risk of developing obesity and other CVD-associated risk factors. Our current results also showed a higher prevalence of obesity in females than in males and the inverse was observed with underweight. Our results concurred with findings of previous studies where obesity was found to be more prevalent among women than among men [21,22].

The current study found a higher value of SBP and DBP in males than in females; this was expected, since males showed higher mean values for both SBP and DBP. Comparable results were reported in Chang et al. [3,23]. The physiological reason for this observed result might be that young women are exposed to the hormones oestrogen and progesterone, which affects the regulation of BP; this may be a reason why men always have high BP [24]. This may also be the reason for the higher risk of HT in a postmenopausal woman since they are no longer exposed to those hormones [25].

Body Mass Index showed a positive significant correlation with the Conicity Index. The results correspond with the findings of Shidfar et al. [26], who found a correlation of BMI with Conicity Index. Pearson correlation also indicated a significant positive correlation between the Conicity Index and FBG, LDL-C, HDL-C, and TG. These suggest the inverse relationship of increase in Conicity Index with lipids profile and FBG. The same results were reported in Andrade et al. [1]. There was no significant correlation between the Conicity Index and BP. Our results contrast with that of Zhou et al. [27], who found a correlation between Conicity Index and BP. 

In our findings, ABSI was only significantly associated with PR. Since ABSI is an indicator of visceral fats, this demonstrates that individuals with increased ABSI have reduced heart rate. Hence, decreased heart rates have been observed in obese people [10]. The current findings correspond with the one of Sowmya et al. [10], who found an association between ABSI and heart rate. In the current study, ABSI was not significantly associated with HOMA−IR; however, Mameli et al. [20] found a significant association between ABSI and HOMA−IR. This contradiction may be due to them having prepared their study on children and adolescents aged 2–18, while in this current study we focused on young adults aged 21–30 years old. 

Body Roundness Index was significantly associated with LDL-C before adjusting the confounding. These results coincide with the results of Snijder et al. [28]. HOMA-IR was not associated with anthropometric indices (Conicity Index, BRI, and ABSI). Insulin resistance together with obesity plays a major role in the development of diabetes [29]. 

The current findings indicate that BRI and Conicity Index were associated with dyslipidaemia. Similar results were reported in Krakauer and Krakauer [30], which supported that ABSI was more associated with premature mortality. Both the BRI and ABSI were first established in Western countries (in North America), and other studies recommended that they must be modified [31,32]. In our findings, the Conicity Index was associated with HT. The ABSI was found to be non-significantly associated with DM2 and systemic arterial HT [1]. In agreement with Fujita et al. [31], ABSI was not a good estimator of HT, dyslipidaemia, and diabetes in Japanese adults. Furthermore, Haghighatdoost et al. [33] found that ABSI was not associated with CVD risk factors. Enough literature supported the association between BRI and dyslipidaemia, and this index can estimate cardiovascular risk factors better [3,7]. Earlier, Sebati et al. [34] reported that the central obesity indices, waist circumference and waist-to-height ratio, were significantly associated with dyslipidaemia and hypercholesterolaemia, whereas Body Mass Index was associated with hypertension among the Ellisras population.

All indices were associated with obesity and underweight. In general, our findings showed associations between selected anthropometric indices and CVD risk factors. These findings demonstrate that Conicity Index, ABSI, and BRI were associated with CVD risk factors among the Ellisras rural young adults. These indices were established as indicators of body fat distribution and their values increase according to the build-up of fats in the abdominal region of the body [4,9]. The relationship of these indices with several CVD risk factors leads us to the reported evidence indicating that body shape, rather than total adiposity determines risk. This demonstrates strong clinical significance in the development of chronic diseases of lifestyle or CVD risk factors [35]. Higher accumulation of fat around the abdomen reduces life expectancy.

There were limitations that were observed in this study. Given the cross-sectional nature of the current study, a causal relationship could not be achieved. We did not consider the family history of cardiovascular diseases, physical activity, alcohol drinking, and smoking pattern, which were reported to have a strong risk factor for future cardiovascular diseases [36,37]. Furthermore, the study sample size was rather small compared to other studies. Finally, all our participants were from the black rural South African population, as such, these findings may not be generalisable to a broader South African population. 

## 5. Conclusions

The findings of our study showed a positive association between BRI, ABSI, and Conicity Index and selected cardiovascular risk factors. BRI was significantly associated with LDLC and dyslipidaemia. ABSI was found to have a positive correlation with PR. Conicity Index was positively associated with insulin resistance, HT, and dyslipidaemia. The study participants were at risk of abdominal obesity and associated cardiovascular risk factors. Further investigation of these indices and their association with nutritional status and cardiovascular diseases (CVDs) could assist in efforts to prevent CVD in the rural South African population.

## Figures and Tables

**Figure 1 ijerph-18-00281-f001:**
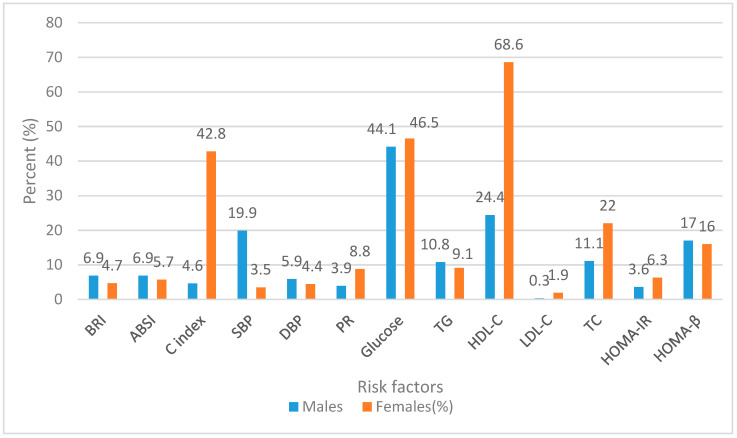
Shows the prevalence of cardiovascular disease (CVD) risk factors using the Body Roundness Index (BRI), A Body Shape Index (ABSI), Conicity Index, blood pressure (BP), and blood samples of Ellisras rural young adults aged 21–30 years old.

**Figure 2 ijerph-18-00281-f002:**
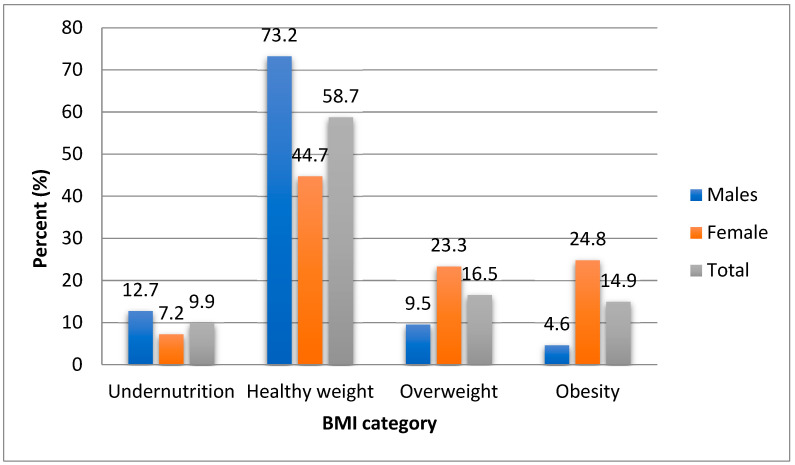
Shows the prevalence of malnutrition in Ellisras young adult males and females aged 21–30 years.

**Table 1 ijerph-18-00281-t001:** Descriptive statistics of anthropometric indices, blood pressure, and biochemical measurements of Ellisras young adults aged 21–30 years.

	SBP (mmHg) M (SD)	DBP (mmHg) M (SD)	PR (bpm) M (SD)	Insulin (mmol/L) M (SD)	C Index M (SD)	ABSI M (SD)	BRI (SD)	BMI (kg/m^2^) M (SD)	TC (mmol/L) M (SD)	Glucose (mmol/L) M (SD)	HDL-C (mmol/L) M (SD)	TG (mmol/L) M (SD)	HOMA-IR M (SD)	HOMA-β M (SD)	LDL-C (mmol/L) M (SD)
**Males**	125.91 (12.48) **	71.44 (10.24) *	70.36 (12.90) **	7.75 (9.00) *	1.12 (0.08) **	1.29 (0.11) **	6.05 (3.96) **	23.67 (8.74) **	2.992 (4.67)	5.45 (0.87)	1.20 (0.37)	1.06 (0.65)	2.29 (3.37)	24.31 (30.14) **	2.35 (0.78) *
**Females**	114.23 (10.84) **	69.11 (9.39) *	81.75 (11.79) **	10.86 (14.04) *	1.16 (0.10) **	1.21 (0.11) **	8.86 (7.99) **	25.66 (17.45) **	3.167 (5.16)	5.62 (1.55)	1.09 (0.30)	0.96 (0.51)	2.48 (3.45)	35.49 (47.33) **	2.73 (0.91) *

M = mean; SD = standard deviation; C index = Conicity Index; ABSI = A Body Shape Index; BRI = Body Roundness Index; BMI = Body Mass Index; TC = total cholesterol; TG = triglycerides; HDL-C = high-density lipoprotein cholesterol; LDL-C = low-density lipoprotein cholesterol; SBP = systolic blood pressure; DBP = diastolic blood pressure; HOMA = homeostasis model assessment; PR = pulse rate; * *p* < 0.05; ** *p* < 0.001.

**Table 2 ijerph-18-00281-t002:** Pearson correlation for anthropometric indices and cardiovascular risk factors of Ellisras young adults aged 21–30 years.

Risk Factors	BRI		ABSI		Conicity Index	
	Males	Females	Males	Females	Males	Females
SBP	−0.009	0.076	−0.035	−0.071	−0.061	0.011
DBP	0.031	0.072	0.001	−0.071	−0.004	0.011
PR	0.078	−0.040	0.148 **	0.053	0.167 **	0.018
Glucose	0.139	0.000	0.084	−0.021	0.141 **	−0.185 **
Insulin	0.252 **	0.255 **	0.036	0.017	0.143 **	0.185 **
HOMA-IR	0.140	−0.040	0.273	−0.020	0.093	−0.018
HOMA-β	0.250 **	0.245 **	0.021	0.012	0.135 **	0.179 **
TC	0.058	0.220 **	0.061	−0.023	0.060	0.125 *
TG	0.310 **	0.216 **	−0.109 *	0.120 *	0.253 **	0.237 **
HDL-C	−0.196 **	0.015	−0.063	0.030	−0.253 **	0.023
LDL-C	0.033	0.113 *	−0.043	0.152 **	0.003	0.183 **
BMI	0.144 **	−0.001	0.091	−0.021	0.150 **	0.016

C index = Conicity Index; ABSI = A Body Shape Index; BRI = Body Roundness Index; TC = total cholesterol; TG = triglycerides; HDL-C = high-density lipoprotein cholesterol; LDL-C = low-density lipoprotein cholesterol; SBP = systolic blood pressure; DBP = diastolic blood pressure; PR = pulse rate; *: *p* < 0.05; **: *p* < 0.001.

**Table 3 ijerph-18-00281-t003:** Linear regression of anthropometric indices and cardiovascular risk factors of Ellisras young adults aged 21–30 years.

Risk Factors	Unadjusted				Adjusted for Age and Gender			
	β	*p*-Value	95% CI		β	*p*-Value	95% CI	
**Conicity Index**								
SBP	−0.075	0.062	0.000	−0.073	−0.033	0.417	−0.054	0.022
DBP	−0.051	0.199	−0.079	0.017	−0.033	0.424	−0.069	0.029
PR	0.099	0.013 *	0.017	0.143	0.081	0.048 *	0.000	0.130
Glucose	0.052	0.190	−0.021	0.106	0.046	0.267	−0.028	0.102
Insulin	0.149	<0.001 **	0.286	0.908	0.110	0.007 *	0.123	0.757
HOMA−IR	0.048	0.236	−0.132	0.536	0.054	0.195	−0.117	0.573
HOMA−β	0.142	<0.001 **	0.259	0.891	0.103	0.011 *	0.096	0.740
TG	0.146	<0.001 **	0.147	0.486	0.143	<0.001 **	0.138	0.482
HDL−C	−0.070	0.081	−0.184	0.011	−0.036	0.381	−0.144	0.055
LDL−C	0.125	0.002 *	0.065	0.284	0.081	0.044 *	0.003	0.225
TC	0.073	0.068	−0.006	0.166	0.040	0.334	−0.045	0.132
BMI	0.063	0.116	−0.019	0.175	0.049	0.238	−0.040	0.161
**ABSI**								
SBP	−0.012	0.770	−0.027	0.037	0.059	0.161	−0.058	0.010
DBP	0.032	0.423	−0.058	0.024	−0.067	0.119	−0.078	0.009
PR	−0.021	0.593	−0.039	0.069	0.081	0.056	−0.001	0.113
Glucose	0.008	0.835	−0.049	0.060	0.033	0.443	−0.035	0.081
Insulin	−0.044	0.271	−0.421	0.118	0.021	0.613	−0.210	0.356
HOMA−IR	−0.004	0.922	−0.273	0.301	0.022	0.612	−0.227	0.385
HOMA−β	−0.047	0.244	−0.435	0.111	0.016	0.709	−0.232	0.341
TG	0.090	0.024 *	0.022	0.314	0.061	0.149	−0.041	0.267
HDL−C	0.057	0.152	−0.023	0.145	0.005	0.912	−0.083	0.093
LDL−C	−0.044	0.272	−0.148	0.042	0.040	0.335	−0.050	0.147
TC	−0.049	0.221	−0.120	0.028	−0.024	0.572	−0.101	0.056
BMI	−0.001	0.986	−0.084	0.083	0.031	0.470	−0.056	0.122
**BRI**								
SBP	0.028	0.481	−0.001	0.000	0.019	0.642	0.000	0.001
DBP	0.113	0.005 *	0.000	0.002	0.142	0.001 *	0.001	.002
PR	0.055	0.169	0.000	0.002	0.028	0.488	−0.001	0.001
Glucose	0.076	0.058	0.000	0.002	0.068	0.097	0.000	0.002
Insulin	0.152	<0.001 **	0.005	0.014	0.113	0.005 *	0.002	0.012
HOMA−IR	0.008	0.848	−0.006	0.005	0.010	0.814	−0.006	0.005
HOMA−β	0.142	<0.001 **	0.004	0.013	0.102	0.012 *	0.001	0.011
TG	0.106	0.008 *	0.001	0.006	0.112	0.006 *	0.001	0.006
HDL−C	−0.045	0.260	−0.002	0.001	−0.007	0.859	−0.002	0.001
LDL−C	0.091	0.023 *	0.000	0.004	0.040	0.315	−0.001	0.003
TC	0.038	0.001 *	0.001	0.004	0.112	0.006 *	0.001	0.003
BMI	0.066	0.100	0.000	0.003	0.050	0.225	−0.001	0.002

*: *p* < 0.05; **: *p* < 0.001 β = beta; C index = Conicity Index; ABSI = A Body Shape Index; BRI = Body Roundness Index; TC = total cholesterol; TG = triglycerides; HDL−C = high-density lipoprotein cholesterol; LDL−C = low-density lipoprotein cholesterol; SBP = systolic blood pressure; DBP = diastolic blood pressure; PR = pulse rate.

**Table 4 ijerph-18-00281-t004:** The odds ratio (OR) and 95% confidence interval (CI) for the association of Conicity Index, BRI, and ABSI values with the prevalence of T2M, HT, and dyslipidaemia in unadjusted and adjusted for age and gender models.

Risk Factors	Unadjusted				Adjusted for Age and Gender			
	OR	*p*-Value	95% CI		OR	*p*-Value	95% CI	
T2M								
ABSI	1.168	0.668	0.574	2.380	1.220	0.587	0.596	2.498
BRI	1.645	0.182	0.792	3.419	1.601	0.213	0.764	3.358
Conicity Index	0.935	0.1841	0.483	1.808	0.891	0.737	0.452	1.754
Dyslipidaemia								
ABSI	0.831	0.654	0.390	1.806	1.013	0.975	0.464	2.207
BRI	2.721	0.007 *	1.307	5.665	2.138	0.047 *	1.010	4.526
Conicity Index	1.945	0.048 *	1.007	3.757	1.391	0.342	0.704	2.748
HT								
ABSI	2.306	0.102	0.846	6.285	1.596	0.377	0.566	4.502
BRI	0.801	0.769	0.185	3.462	1.568	0.568	0.332	7.459
Conicity Index	1.808	0.620	0.174	18.807	3.985	0.029 *	1.395	4.522
IR								
ABSI	0.434	0.595	0.020	9.404	0.881	0.942	0.029	26.659
BRI	1.028	0.071	0.998	1.058	1.023	0.135	0.993	1.054
Conicity Index	7.761	0.001 **	5.783	96.442	4.646	0.007 *	2.792	74.331
Beta−cell dysfunction								
ABSI	1.855	0.688	0.091	37.784	2.336	0.600	0.098	55.554
BRI	1.104	0.143	0.967	1.260	1.117	0.141	0.091	1.294
Conicity Index	4.418	0.101	0.594	34.652	5.722	0.103	0.575	43.150
Underweight								
ABSI	6.537	<0.001 **	7.211	10.746	6.533	0.002 *	4.414	85.746
BRI	0.201	<0.001 **	0.127	0.316	1.600	0.157	0.834	3.067
Conicity Index	0.023	<0.001 **	0.251	0.433	0.031	<0.001 **	0.411	0.612
Obesity								
ABSI	0.019	<0.021 *	0.004	0.095	0.123	0.016 *	0.022	0.675
BRI	3.607	<0.001 **	2.911	4.471	3.557	<0.001 **	2.841	4.454
Conicity Index	1.058	<0.001 **	2. 715	4.119	1.271	<0.001 **	0.672	1.099

*: *p* < 0.05; **: *p* < 0.001; T2M = type 2 diabetes; HT = hypertension; IR = insulin resistance; ABSI = A Body Shape Index; BRI = Body Roundness Index.

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
