# Peer review of "Body Roundness Index, A Body Shape Index, Conicity Index, and Their Association with Nutritional Status and Cardiovascular Risk Factors in South African Rural Young Adults"

_ijerph, 2021, doi:10.3390/ijerph18010281_

Round 1

Reviewer 1 Report

Mbelege et al studied the association of body roundness index, a body shape index, and conicity index with nutritional status and cardiovascular risk factors in south African rural young adults. Conicity index can predict the chances of insulin resistance, hypertension, and dyslipidemia. This meta-analysis is an interesting study. There are several issues the authors need to address.

  1. The author using number of participants as Y axis is meaningless in Figure 1. The point is comparing indicate high prevalence for BRI, ABSI, SBP and DBP between males and females.
  2. The author should show positive or negative association in the conclusion or summary sentences.
  3. In page 2, line 48, BMI unit should be kg/m2
  4. For the Pearson correlation analyses, they should show the r and p values.
  5. P value should be detailed not 0.000
  6. Initial casing rule is confusing in many places.
  7. The English writing need improve.

Author Response

Reviwer1

Mbelege et al studied the association of body roundness index, a body shape index, and conicity index with nutritional status and cardiovascular risk factors in south African rural young adults. Conicity index can predict the chances of insulin resistance, hypertension, and dyslipidemia. This meta-analysis is an interesting study. There are several issues the authors need to address.

1,using number of participants as Y axis is meaningless in figure 1.the point is compering indicate high prevalence for BRI,ABSI,SBP  and  between males and females DBP.

Response: Thank you for your useful comment the, the Y axis which was number of participants was amended to percentages  in figure 1

2,the author should show positive or negative association in conclusion or summary sentences

Response: Thank you for your comment, the suggestions effected as suggested.  . See line 281.

3,in page 2,line 48,BMI unit should be kg/m2

Response: Changes have been implemented as  suggested. See line 36.

4,for the pearson correlation analysis,they should show the r and p values

Response: Thank you for the comments. The changes effected as suggested. See line 180

5,the  p value should be detailed not  0.000

Response: Thank you for your comment, Changes effected as suggested. See line 185-191.

6,initial causing rule is confusing in many places

Response: The sentence has been deleted. In addressing the causing rule please see limitation of the study line 273.

7, the English writing need improve

Response: Thank you for your comment, the manuscript was sent to a professional native  English language editor.

Reviewer 2 Report

This study examines the association of various body types with CV risk factors. 

While data is of interest "there are too many trees, and no forest". There is a plethora of details but no "big picture". What do the results mean? interpret them! After doing numerous analyses what message do you take away? A long list of correlations is difficult to read and leaves the reader saying "so what?"

Please make the DISCUSSION a "big picture" interpretation of your results. Also are your results any better than BMI and WC?

ADDITIONAL COMMENTS

Finally lines 1-45 on the INTRO can be removed.

Also in one of you tables you write that values are associated with the "development" of diabetes, HTN, etc. This is a cross sectional paper. It should be "prevalence" of disorders.

Author Response

Reviewer 2

Comments and suggestions to authors

The study examine the association of various body types with CVD risk factors.

while data is of interest ” there are too many trees and forest”. There is a plethora of details but “no big picture”. What do the results mean? Interpret them! A long list of correlations is difficult to read and leaves the reader saying “so what”

Please make the Discussion a big picture interpretation of your results.

Response: Thank you for your useful comment, Suggestion is effected as suggested. See line 264 to 272

After doing numerous analyses what message do you take away?

Response: Thank you for the comments. The comments effected as suggested. See line 266 to 270

 Also are your results any better than BMI and WC

Response: Thanks you for the comments. The changes effected as suggested. See line 261  to 263. That “Earlier, Sebati et al [35] reported that the central obesity indices, waist circumference and waist to height ratio, are better predictors of dyslipedemia and hypercholesterolemia, whereas body mass index was a better predictor of hypertension among these Ellisras population.”

Additional comments

-Finally 1-45 on the intro can be removed.

Response: Thank you for your comment, Comments effected as suggested. See introduction section.

Also in one of your tables you write that values are associated with the development of diabetes, HTN etc. this is a cross sectional paper.it should be prevalence of disorders.

Response: Amended, replaced with “the prevalence” in table 4.

Reviewer 3 Report

This is a well planned, conducted and described study.
In my opinion, it deserves to be published after appropriate correction.
From my perspective, there is a lack of smoking among the cardiovascular risk factors considered. Obtaining smoking data, e.g. using the survey method, is simple. The omission of smoking as a risk factor for CVD is difficult to understand. The manuscript should be supplemented with such data.
An interesting addition would also be to consider the level of physical activity as a factor associated with cardiovascular risk.
Authors should discuss the limitations of the study.
Some of the oldest references may be omitted.

Author Response

Reviewer 3

Comments and suggestions

This is a well planned ,conducted and described study.in my opinion, it deserves to be published after appropriate correction.

From my perspective, there is lack of smocking among the cardiovascular risk factors considered. Obtaining smoking data, e.g using the survey method, is simple. The omission of smoking as a risk factor for  CVD is  difficult to understand. The manuscript should be      supplemented with such data.

An interesting addition would also be consider the level of physical activity as a factors associated with cardiovascular risk.

 Response: Thank you for your suggestion, indeed smoking and physical activity data would be of useful to this study as risk factor of cardiovascular diseases. However, the such data was not available for analysis. See limitation of the study Line 273-278

-Author should discuss the limitation of the study.

Response: Thank you for your  useful comment, the limitation of the study were discussed. See line 273-274 to 281

-Some of the oldest references may be omitted.

Response: Thank you for your comments. The comments effected as suggested. However, the main sources for the standard/method use were maintained. These are articles that carried out the original methodology of the study.

Round 2

Reviewer 1 Report

Fine

Author Response

Thank you

Reviewer 2 Report

The paper is improved but it still needs more revisions:

  1. Line 1 of INTRO - rewrite. It makes no sense
  2. RESULTS - remove all the numbers from the text. Just say factors x, y and z were significantly related to shape A or B or C. The actual values are in the figures or tables. As it stands now, reading the text is tortuous and difficult.
  3. In the DISC section and CONCLUSION sections the authors discuss predicting CVD or mortality. Remove this and find alternate wording. Cross sectional studies do not predict.
  4. The start of the DISC section should be changed. "In this x-sectional study of rural Africans we found that body shape X had the strongest associations with CVD risk factors, while shape Y had the weakest associations. Shape Z had intermediate associations." This way you summarize your results in a succinct manner. You need to give a broad over-view of the results.
  5. You need to define the study cohort. I assume they are native Africans and they are rural. Hence the results are unique to them and not necessarily applicable to whites or to urban Africans. This needs to be emphasized.
  6. In Table 4, under obesity, the ABSI odds ratio is close to zero and the p value is <0.001. This does not make sense. Please check

Author Response

Comments and Suggestions for Authors

The paper is improved but it still needs more revisions:

  1. Line 1 of INTRO - rewrite. It makes no sense

Response: thank you for your comment the sentence was rephrased.

  1. RESULTS - remove all the numbers from the text. Just say factors x, y and z were significantly related to shape A or B or C. The actual values are in the figures or tables. As it stands now, reading the text is tortuous and difficult.

Response; thank you for your comment. The suggestion effected as suggested by the subject editor.

  1. In the DISC section and CONCLUSION sections the authors discuss predicting CVD or mortality. Remove this and find alternate wording. Cross sectional studies do not predict.

Response; thank you for your useful comment, the predicting word was removed and replaced with association.

  1. The start of the DISC section should be changed. "In this x-sectional study of rural Africans we found that body shape X had the strongest associations with CVD risk factors, while shape Y had the weakest associations. Shape Z had intermediate associations." This way you summarize your results in a succinct manner. You need to give a broad over-view of the results.

Response: thank you for your useful suggestion, the start of discussion was amended.

  1. You need to define the study cohort. I assume they are native Africans and they are rural. Hence the results are unique to them and not necessarily applicable to whites or to urban Africans. This needs to be emphasized.

Response: thank you for your comment, yes the participants are only blacks from rural area and the point was emphasized on line 227 under limitations.

  1. In Table 4, under obesity, the ABSI odds ratio is close to zero and the p value

is <0.001. This does not make sense. Please check

Response: thank you for your comment, The results were re-checked and corrected

Reviewer 3 Report

Thank you for responding to my comments.

The current version of the article is acceptable.

Author Response

Thank you